# Visual Tracking via Deep Feature Fusion and Correlation Filters

**DOI:** 10.3390/s20123370

**Published:** 2020-06-14

**Authors:** Haoran Xia, Yuanping Zhang, Ming Yang, Yufang Zhao

**Affiliations:** 1College of Computer and Information Science, Southwest University, Chongqing 400715, China; hrzq7126z@email.swu.edu.cn (H.X.); yangming@swu.edu.cn (M.Y.); 2Faculty of Psychology, Southwest University, Chongqing 400715, China; zhaobee@swu.edu.cn

**Keywords:** visual taracking, convolution neural networks, corerelation filters

## Abstract

Visual tracking is a fundamental vision task that tries to figure out instances of several object classes from videos and images. It has attracted much attention for providing the basic semantic information for numerous applications. Over the past 10 years, visual tracking has made a great progress, but huge challenges still exist in many real-world applications. The facade of a target can be transformed significantly by pose changing, occlusion, and sudden movement, which possibly leads to a sudden target loss. This paper builds a hybrid tracker combining the deep feature method and correlation filter to solve this challenge, and verifies its powerful characteristics. Specifically, an effective visual tracking method is proposed to address the problem of low tracking accuracy due to the limitations of traditional artificial feature models, then rich hiearchical features of Convolutional Neural Networks are used to make the multi-layer features fusion improve the tracker learning accuracy. Finally, a large number of experiments are conducted on benchmark data sets OBT-100 and OBT-50, and show that our proposed algorithm is effective.

## 1. Introduction

With the development of artificial intelligence and computer vision system, visual tracking has become more and more important in the field of computer vision applications. It has been widely used in surveillance, driver assistance system, remote sensing, defense system, or human–computer interaction. Convolution Neural Network (known as CNN) based feature extraction have also been successfully used to the multi-target tracking problem [1,2,3,4,5], and there are many good results [6,7,8,9]. Although existing tracking algorithms have achieved good performances, it is still a challenge because of the uncertainty and variable factors of target objects, such as occlusion, illumination, scale changes and attitude changes.

Normally, positive training samples and negative training samples are collected near the estimated target position by some existing deep-learning-based trackers [10,11,12,13], and the classifier is gradually trained by extracting features from CNN. All of this raises two questions, first of all, most target recognition algorithms use neural networks as online classifiers, and only use the output of the last layer as targets, which is conducive to express the semantics of the targets. However, the main goal of visual tracking is to pinpoint the target location, not to infer the semantic of the target. Therefore, using only the feature of the last CNN layer does not work well on tracking. The second question is how to extract training samples. A large number of diverse samples are needed to train a robust neural network classifier, yet it is not easy to implement in real-time visual tracking system. To solve these questions, we apply adaptive correlation filtering to the features which are extracted by each layer of CNN, and also use it to co-locate and infer the target position. We have worked in this paper along the following three lines.

Based on previous researches, we find that using the last layer of CNN can extract effective semantic information but not fine-grained spatial information, for example, the object location. On the other hand, the front-end layer of CNN is accurately positioned but not used to extract object semantics. Therefore, we use the rich feature levels of CNN as the target representation of visual tracking, and at the same time using semantic and fine-grained details to deal with the huge changes in appearance to avoid the drift and loss of targetsUsing the search strategy from rough to fine and the hierarchical characteristics of CNN research and the multi-level reasoning mechanism of classical computer vision to obtain the optimal tracking results, at the same time combining the correlation filter on each CNN layer to conduct adaptive learning while the multi-stage correlation filter is used to implement co-positioning, and make the target position be inferred from rough to fine [14].A plenty of experiments on large data sets which called OBT-100 [15] are tested, and the results of experiments show that our tracking algorithm is better than the existing methods in terms of accuracy and robustness.

## 2. Related Work

### 2.1. Original Detection Tracking

We can consider visual tracking [10,11,16] (also known as detection tracking) as the process of repeatedly detecting the location of the target within a window, often using online learning methods to classify the target objects. However, this method will bring small errors in the sample mark. With the gradual accumulation of errors, the tracking of the target will drift. This is commonly referred as the sample uncertainty problem. In order to solve the problem of sample uncertainty and subsequent model updating, some fuzzy-based tracking methods are proposed. Ramirez-Paredes et al. [17] proposed a target tracking method based on fuzzy inference, which can be used to automatically and selectively update the target tracking of existing models. Kim et al. [18] applied target segmentation and tracking based on fuzzy membership distribution. Li et al. [19] adopted a fuzzy logic data track-to-track association approach for online visual multi-target tracking and solve the problem of track fragment splicing effectively. Of course, there are many other methods and they are similar in that they aim to update the classifier correctly and efficiently to reduce drift, such as multi-instance learning [20], semi-supervised learning [21], Tracking with Gaussian Processes Regression (TGPR) method [22] and Tracking-Learning-Detection (TLD) framework with P-N learning [23]. Multiple Experts using Entropy Minimization (MEEM) method [24] learned several classifiers at different learning rates and used them in combination, achieving a good tracking effect. In addition, Hare et al. [25] combined the target tracking of accurate position estimation with attitude tracking, and used its multi-level structured output for prediction, thus reducing the impact of sample uncertainty on the experiment, which performed well in the recent benchmark study [26]. In this paper, we also use correlation filters to solve such problems, but we make the training samples return to the soft labeling of Gaussian function instead of binary labeling to learn the discriminative classifier.

### 2.2. Tracking by Convolutional Neural Networks

Similar methods, such as incremental learning [27] and Differential Earth Mover’s Distance (DEMD) algorithm [28], are representative of the appearance characteristics of objects using handcrafted features. In recent years, CNNs have greatly promoted the development of visual recognition. Deep learning tracker (DLT) [10] proposed by Wang, used multi-layer automatic encoders. In addition, Wang [11] set up a neural network with two layers of continuous learning and constrained the learning process by time. Li [12] used multiple CNN classifiers to act on different instances of the same target object to eliminate the influence of noise samples in the process of model updating. These CNN trackers typically only take advantage of the last layer of CNN. So we want to combine the multi-layer structure of CNN to represent the target characteristics in order to effectively reduce the uncertainty of sampling.

### 2.3. Tracking by Correlation Filter and Multiple Feature Fusion Strategy

Bolme [29] first introduced relevant filters into visual tracking named Minimum Output Sum of Squared Error (MOSSE) filter, and then on this basis, Circulant Structure of Tracking-by-Detection with Kernels (CSK) [30], Kernelized Correlation Filters method (KCF) [31] and Color Name (CN) [32] methods were proposed. Inspired by MOSSE and CSK methods, Zhang [33] proposed the tracking algorithm STC, it takes the context information around the target as an important reference to make the target position more accurate. The Superpixel Tracking algorithm (LCT) [34] decomposed the tracking task into two steps of target location and scale estimation, and combined the context information in the tracking, and used the tracking results to train the online randomization classifier to realize the redetection of failed targets, so as to ensure the long-term tracking. In order to obtain a single-channel response that can mark the position of an object, Ruan proposed a correlation filter (MVCF) [35] that can convolve directly with a multi-vector descriptor. Ma [36] used the multilayer abstraction of image pyramid to represent the target object, and achieved good results in the large-scale test set. However, this algorithm ignores the importance of handcrafted features in object tracking in complex scenarios, and it is also very time consuming to infer from rough to fine. Most of the existing feature selected tracking algorithms only use a single fusion strategy to merge traditional features or depth features, which results in the weak robustness of the algorithm, and the tracking is vulnerable to the impact of environmental changes. Histogram of Oriented Gradient (HOG) [37] feature is a descriptor that can quickly describe the local gradient feature of an object. For the purpose of making visual tracking more robust, we proposes a correlation filtering object tracking algorithm that integrates with the HOG feature and the CNN layered feature. In addition, inspired by the other improved algorithms mentioned above, this paper also adopts region proposal for scale estimation to solve the problem of target scale change and slow speed caused by the combination of multiple features. In addition, the real-time performance and stability of the tracking algorithm are improved.

## 3. Overview

In Figure 1, we show a visual example of the CNN hierarchical characteristics of an image with edges. The schematic diagram of the tracking algorithm we proposed is shown in Figure 2. For the displacement estimation, we use the features of CNN’s multiple convolutional layers to train the correlation filter.

In the tracking process, when we input a new image, we first cut out the search window of the previous frame’s estimated position and input it into the CNN network. The feature output of each convolutional layer of CNNs is then taken as the target object, and the convolution operation is performed again with the learned correlation filter to generate the response graph. The maximum confidence value represents the target position. Next, a coarse-to-fine multi-level response mapping search is performed. The obtained search results are weighted with the HOG feature response graph to infer the location of the target. Long-term memory filter $Z_{L}$ is used to calculate the confidence coefficient *g* of the inputted image area, and check whether the confidence coefficient is lower than the given threshold value $T_{r}$ to detect if the target tracking fails. We generate candidate regions in the entire input image by Region Proposal. If the object is lost, we use $Z_{L}$ to calculate the confidence coefficient of these candidate regions, and then take the highest confidence as the result of retesting. On the scale estimation of target, we generate a candidate region with a small step size to make them closely around the estimated target position.

## 4. Proposed Methods

This section elaborates on the method of object tracking proposed in this paper, including region proposal method, convolutional layer feature extraction, adaptive feature fusion, correlation filter and adaptive feature weighted fusion method. On this basis, we propose the method of target weight detection and target scale estimation based on the candidate region. Finally, we introduce the methods of updating correlation filters and memory filters at different learning rates.

### 4.1. Region Proposal

An important process of target tracking is to select the optimal target area and redetect the target object when the tracking fails. In this process, we used the method similar to EdgeBox [38] to select two candidate regions, namely, the region proposal $B_{s}$ with a small step size that is close to the target, and the $B_{d}$ with a large step size that may span the entire image range. The region rejection technique proposed by Li et al. [39] was then used to select the region and filter the non-optimal terms. Finally, the moving average algorithm is used to update the target. This work can also be divided into two parts: the selection of the region proposal center, the optimization about proposal length and width. For each region proposal, also called the candidate bounding box, let (x,y) be the center coordinate and (w,h) be the height and width. During the process of calculating the confidence score of each bounding box, we use the correlation filter $Z_{L}$ to learn the subtle scale changes of the bounding box on the conv3-4 layers of CNN. g(b) represents the maximum response value of the long-term memory correlation filter.

Let $\left(x_{t},y_{t}\right)$ be the center of the image at frame *t*. So we are going to center this point as region *z*, and make the size of region *z* consistent with the previous frame. Let $T_{r}$ be a threshold that determines whether a trace failure has occurred or not. When confidence g(z) is lower than $T_{r}$, we mark the tracker as the target lost state and begin the target redetection process. Instead of simply selecting the most credible candidate bounding box as the result of the recovery, we consider the drastic changes that possibly occur between successive frames. Calculate the distance D between the center point of each proposal *b* from frame t−1 to frame *t*, *i* is the serial number of each proposal:(1)$$D(b{}_{t}^{i},\;b{}_{t-1}^{i})\;=\;exp(-\frac{1}{2\sigma {}^{2}}(∥\left(x{}_{t}^{i},\;y{}_{t}^{i})-(x{}_{t-1}^{i},\;y{}_{t-1}^{i}\right)∥{}^{2}))$$
where σ is used to indicate the initial target size value of the diagonal length. In order to select the optimal solution, we solve the following problems by finding the minimum value, the distance D between the center positions is obtained by Equation (Equation 1):(2)$$\begin{array}{cc} \; & \underset{i}{\arg\min}\;\left\{g\left(b{}_{t}^{i}\right)+\alpha D\left(b{}_{t}^{i},b{}_{t-1}^{i}\right)\right\}\\ \; & s.t.g\left(b{}_{t}^{i}\right)\geq 1.5T{}_{r} \end{array}$$
where α is the weighting factor to achieve a balance between image area confidence coefficient and smoothness of motion.

In combination with the EdgeBox [38] method mentioned earlier, the scale estimation was carried out, and the multi-scale region was generated by the minimum step size and the minimum non-maximum inhibition (NMS) threshold. The proposal rejection technique [39] was then used to filter out proposals whose Intersection over Union (IoU) value was less than 0.6 or greater than 0.9. IoU value can be calculate by the estimated target position $b_{1}$ and the target position $b_{0}$ of ground-truth, such as $IoU=\frac{\left|b{}_{1}\cap b{}_{0}\right|}{\left|b{}_{1}\cup b{}_{0}\right|}\geq 0.5$. Finally, the resulting proposal was set to a fixed value to calculate the confidence coefficient score. If the max confidence $g\left(b\right)|b\in B_{s}$ was greater than g(z), then we used the moving average algorithm to update the width and height of the target:(3)$$\left(w{}_{t},\;h{}_{t}\right)\;=\;\eta \left(w{}_{t}^{*},\;h{}_{t}^{*}\right)\;+\;\left(1-\eta \right)\left(w{}_{t-1},\;h{}_{t-1}\right)$$
where $\left(w{}_{t}^{*},\;h{}_{t}^{*}\right)$ represents the width and height of the maximum confidence coefficient proposal, and the weight factor η makes the target scale estimation change smoothly.

### 4.2. Convolution Layer Feature Extraction

Since the birth of convolutional neural network, image processing technology has developed rapidly, and great progress has been made in each subdivision of the research field. It can be said that CNN is currently the best partner for image processing. We convolve an image to get a set of features that can be further convolved, and so on, to get a unique, and fully connected feature. In CNN, we have an input layer, a convolution layer, a pooling layer and a fully connected layer. The data of the input image is extracted through multiple convolution layers and a pooling layer, which is gradually transformed from low-level features to high-level features.Then the final features are classified according to the full connection layer, and the image semantics are represented by a very long vector, and the image categories are displayed. So far, CNN technology has become very mature. A variety of network variants of CNN have been emerging, and are widely used in target detection, attitude estimation, face recognition, behavior recognition, semantic segmentation, crowd density estimation, video classification, image quality evaluation and other fields.

We used VGG-Net [40] to extract the CNN feature map to represent the target appearance. Its network configuration is shown in Figure 3. There are six different structures of the network from left to right, representing different neural network models with increasing depth, representing the depth of the model from top to bottom. Each model contained 5 sets of convolution, each set of convolution used 3×3 convolution kernel, and for each convolution done, a 2×2 maximum pooling operation was performed, followed by three full connection layers and soft-max layers. If VGG-Net needed to train the high-level network, it could start from the low-level network and initialize the high-level network with the weight obtained by the former, which could make network convergence speed faster. In CNN, as the features spread to a deeper level, the semantic differences between different types of images were strengthened, but the spatial resolution was gradually reduced. In the process of studying the algorithm of visual object tracking, the main experimental purpose was to obtain the target position, so we were less concerned with its semantic information, and the 1×1 pixel basically could not contain spatial resolution information. We removed three full connection layers in VGG-Net, so as to make our tracking system more lightweight.

The spatial resolution in the convolutional network decreased gradually with the convolutional layer which was getting deeper, because pooling operation was used in CNNs. For example, in VGG-Net, the convolution feature map space size of pool5 was 7×7. This was $\frac{1}{32}$ of the input image which size was 224×224. Such a low spatial resolution was not enough to accurately locate the target, so we used bilinear interpolation to alleviate this problem, adjusting the fixed size of each feature map to make it a larger size. Set *h* and *p* as feature map and up-sampling map, then the position feature vector *p* for position *i* is:(4)$$p{}_{i\;}=\;\sum_{k}a{}_{ik}h{}_{k}$$
where the interpolation parameter $a_{ik}$ depends on the positions of the adjacent eigenvectors *i* and *k*. This interpolation occurs in space and can be regarded as the interpolation of positions. We used the pre-trained VGG-Net-19 [40] as our feature extractor to visualize the upper sampling outputs of conv 3-4, conv 4-4 and conv 5-4 layers of the neural network. As shown in Figure 1, for each convolutional layer, we visualized the first three components of the convolutional channel as RGB values. Over time, the edges in the input graph significantly changed in appearance. In different frames, the fifth convolutional feature of the target border was more or less dark red. Although the background has changed dramatically, it was still different from the background area. This feature of conv 5-4 allowed us to handle significant cosmetic changes and achieve accurate target positioning at a coarse-grained level. In contrast, conv 4-4 and conv 3-4 layers had more spatial details that helps to pinpoint targets at a fine-grained level. Let us illustrate the advantages of using pre-trained CNNs as feature extractors here. Firstly, the pre-trained CNN did not need model adjustment, so the features extracted by CNN were simple and efficient. Secondly, because this CNN model was obtained by training from 1000 object categories and had been trained many times, it could capture a target well which had not been learned in advance, which was convenient for model-free tracking.

### 4.3. Convolution Target Location

The basic idea of a typical tracker based on correlation filtering [29,30,32,41,42] is learning a discriminant classifier firstly, then searching for the maximum value of the relevant response graph to estimate the assumed position of a target object. Our algorithm uses multiple convolutional layers to form multi-channel features to estimate the target location [31,43,44]. We use *q* to represent the eigenvectors of layer *l* whose size is W×H×C, where *W*, *H* and *C* respectively represent the width, height and channel number. Here, we simply represent $q^{\left(l\right)}$ as *q*, ignoring the dependency of the layer index *l* on *W*, *H*, and *C*. Let *q* be looped over *W* and *H* be a learning sample. Each cyclic shift sample is:(5)$$q{}_{w,h},\;\left(w,\;h\right)\in \;\left\{0,\;1,\;\cdots ,\;W-1\right\}\times \left\{0,\;1,\;\cdots ,\;H-1\right\}{}_{}$$
and its Gaussian function is labeled as:(6)$$y{}_{w,h}\;=\;exp\left(-\frac{\left(w-W/2\right){}^{2}+\left(h-H/2\right){}^{2}}{2\sigma {}^{2}}\right)$$

Among them, σ is the kernel bandwidth, and then by minimizing the following questions we can get the correlation filter *z* same as *q* size:(7)$$z^{*}=\underset{z}{\arg\min}\;\sum_{w,h}∥z\cdot q{}_{w,h}\;-y{}_{w,h}∥{}^{2}+\lambda ∥z∥{}^{2}$$
where λ is the regularization parameter (λ≥0), inner product is derived from the linear kernel in the Hilbert space by: $Z\cdot Q{}_{w,h}=\sum_{d=1}^{D}Z{}_{w,h,c}^{T}\;Q{}_{w,h,c}$. As $y_{w,h}$ is a soft label, a hard threshold sample is not required. It is worth noting that the minimization problem in the formula is closely related to the training-related filter in Reference [44], so the fast Fourier transform method (FFT) can be used to find the optimal solution for each channel. The content of using FFT is introduced below. *Q* and *Y* are Fourier transform signals, and the correlation filter in the frequency domain where the $d\left(d\in \left\{1,\cdots ,D\right\}\right)$ channel is located can be expressed as:(8)$$Z{}^{d}=\frac{Y⊙\bar{Q}{}^{d}}{\overset{D}{\underset{i=1}{\sum }}Q{}^{i}⊙\bar{Q}{}^{i}+\lambda }$$
where *Y* is derived from the Fourier transform of
(9)$$y=\left\{y\left(w,h\right)|\left(w,h)\in \left\{0,1,\cdots ,W-1\right\}\times \left\{0,1,\cdots ,H-1\right\}\right.\right\}$$
$\bar{Q}$ is complex conjugate of Q. ⊙ is the element wise product (Hadamard product). Consider the region of the image for any frame, *U* represent the feature vector layer *l*, whose size is W×H×C, and then the relevant response diagram of layer *l* is calculated as followed [29]:(10)$$f\;=\;F{}^{-1}\left(\sum_{d=1}^{D}Z{}^{d}⊙\bar{U}{}^{d}\right)$$

We can estimate the target location of layer *l* by searching the max position of the correlation response graph with size W×C.

### 4.4. Adaptive Convolution Feature Fusion

In CNN, when the convolution reaches a higher level, the semantic information is relatively rich, so it is often used to represent the appearance characteristics and classification of the object. Therefore, its fusion weight needs to be set to a higher value, and the convolutional neural network at low levels make the target boundary obvious. The target can be accurately positioned, but it is sensitive to occlusion deformation and should be given a lower fusion weight. According to the above principle, the weighted sum of the response graph of each layer is used to obtain the final convolution feature response graph $f_{conv}$, that is
(11)$$f{}_{conv}=\sum_{l=1}^{5}a{}_{l}f{}_{l}$$
where $a_{l}$ represents the fusion weight of each layer. Let the weight of the convolution layer from coarse to fine be 0.5, 0.3 and 0.2 respectively.

HOG feature is a traditional feature. Compared with convolution feature, the target representation of HOG in complex scenes is not rich enough. In order to make full use of traditional features and convolution feature, our algorithm makes adaptive weighted fusion of HOG traditional feature response graph $f_{trad}$ and volume feature response graph $f_{conv}$ obtain the final target response graph.

In order to calculate the discrimination ability of traditional features and convolution features in the tracking process, the algorithm uses the difference value of feature response graph Peak Side Lobe Ratio (PSLR) of adjacent image frames to calculate the weight of feature fusion [45]. The smaller the difference is, the better the performance of the filter learned under the feature is, the higher the weight should be assigned to the feature response graph in the fusion of the feature response graph. Setting t as the sequence number of the current frame, the adaptive weight calculation formulas of the two features are these:(12)$$k{}_{t}^{conv}=\frac{C{}_{t}^{trad}}{kC{}_{t}^{conv}+C{}_{t}^{trad}},\;k{}_{t}^{trad}=1-k{}_{t}^{conv}$$
where, $C^{trad}$ and $C^{conv}$ represent the traditional feature of adjacent frames and the peak sidelobe ratio difference of convolution feature response graph respectively. The formula is this:(13)$$C{}_{t}=PSLR\left(f{}_{t}\right)-PSLR\left(f{}_{t-1}\right)\;PSLR\left(f\right)=\frac{\max(f)-\mu }{\sigma }$$
where μ and σ are the mean value and variance of response graph *f* respectively.

Figure 4 is the feature weight change diagram of the multi-feature adaptive fusion method running on the Bolt video sequence. As shown in the figure, the convolution feature weight is large in most frames of the video sequence. Especially when background clutter environment has serious interference to a target, such as Bolt’s 137th frame video sequences, the other athletes clothes and the color of the billboards interference makes traditional feature weight decrease. However, when the target moves fast making the boundary contour fuzzy or meet with similar goals, convolution feature weight slightly diminishes, such as Bolt video sequences of 65 frames and frame 107.

### 4.5. Appearance Model Update

As described in [44], the optimal filter at layer *l* can be updated by the minimum output error of all trace results. However, the method has to solve a set of D×D linear equations at each (m,n) position. The number of CNN channels is frequently large, which consumes a lot of computing resources. For example, VGG-Net makes D=512 in layer conv 5-4 and conv 4-4, so we use the moving average method to update the correlation filter $Z^{d}$ in Equation (Equation 8) to obtain a robust approximation.
(14)$$Z{}_{t}^{d}=\left(1-\eta \right)\;Z{}_{t-1}^{d}+\frac{\eta Y⊙\bar{X{}_{t}}{}^{d}}{\overset{D}{\underset{i=1}{\sum }}X{}_{t}^{i}⊙\bar{X{}_{t}}{}^{i}+\lambda }$$
where η is learning rate.

### 4.6. Implementation Details

The main steps and implementation details of the tracking algorithm are given in Algorithm 1. This algorithm uses VGG-Net-19 [40] trained on ImageNet [46] for feature extraction. After removing the full connection layer, the output of 3-4, 4-4 and 5-4 CNN layers are used to represent feature. The reason why we do not use the output of the pooling layer is that we would like to get more information on convolution layers that can tell the boundary apart. Set the search box size of an image to W×H (1.8 times the target size), then we make a fixed spatial size of W/4×H/4 to adjust the feature channel size of all convolutional layers.

Keep the parameters of each layer of training related filter unchanged. We set the regularization parameter to $\lambda =10^{-4}$ of Equation (Equation 7) and make 0.1 kernel bandwidth to generate Gaussian function labels. Set the learning rate η to 0.01, in order to solve the problem of boundary discontinuity, and channels extracted from all CNN layers are weighted through a cosine window [29]. 3-4, 4-4 and 5-4 CNN layer regularization terms γ is set to 1, 0.5 and 0.2 apart. The initial values of the fusion weights $k^{trad}$ and $k^{conv}$ are set to 0.5. The experiment shows that the target location result is insensitive to the parameter *r* of the constraint condition of neighborhood search. This is due to the use of the method of estimating the position of the target by summing up the multi-layer weighted response graph.

**Algorithm 1:** Tracking algorithm with CNN and correlation filters

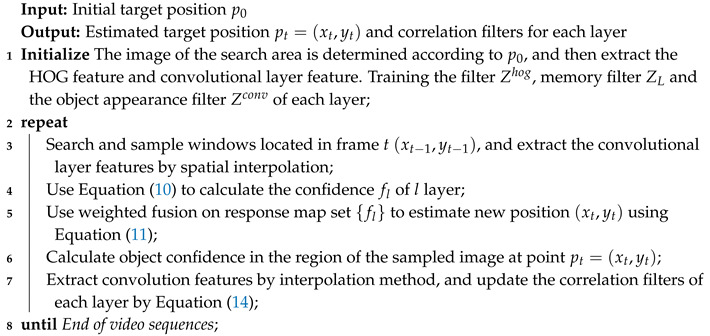



## 5. Experiments

We evaluated the proposed approach on a large benchmark data set OTB-100 [15] and compared it with other recent approaches. To ensure the integrity of the experiment, we also presented the evaluation results on the benchmark data set OTB-50 [47]. In this paper, distance precision, overlap success rate and center location error rate were used to quantitatively grade the tracker. Following the evaluation protocol proposed by [15], we used the same parameter values for all sequences and all sensitivity analyses. We implemented the tracking algorithm proposed in this paper by using MatConvNet toolbox [48] in MATLAB environment, and the forward propagation calculation on CNNs was run on our experiment platform with GeForce GTX Titan GPU and Intel i7-4770 3.40 ghz CPU and 32 GB RAM.

### 5.1. Evaluation Metrics

We used the following latest evaluation criteria of object tracking system to evaluate the proposed methods.

#### 5.1.1. Overlap Success Rate (OS)

The overlap rate between the target position $b_{1}$ estimated by the tracking algorithm and the target position $b_{0}$ of ground-truth which is greater than the frame percentage of the given measure parameter. A given measure parameter is also called Intersection over Union (IoU). We compared the OS value obtained in each frame with the threshold set in advance. When the OS is less than the threshold, we believe that the tracking of this frame has lost the target, otherwise the tracking is successful. The ratio of the number of successful frames to the total number of video frames is the Success Rate. OS values range from 0 to 1, so we can plot a curve.

#### 5.1.2. Distance Precision (DP)

The degree of object offset has a great relationship with the distance between the center points (ground-truth) before and after its movement (Bounding Box). If the actual distance is within the range of the set threshold, the experimental results are not bad. Different thresholds may result in different DP rates, so we can draw a curve of experimental data that contains obvious changes. The disadvantage of this method is that when the target is zoomed or the size changes, it is not very comprehensive to use the center point as the index.

#### 5.1.3. Center Location Error (CLE)

This indicator is complementary to the previous one. The average Euclidean distance has its own advantages on this issue.

#### 5.1.4. One-Pass Evaluation (OPE)

All of our average accuracy and success rate are based on the first frame, and the initialization of the first frame is achieved by ground-truth. There are two disadvantages to this approach. First of all, the tracking algorithm may be sensitive to the initial position given in frame 1, which will has a great impact on the accuracy of the initial position. Secondly, most algorithms do not have the mechanism to reinitialize after the tracking failure, which will seriously affect the tracking effect.

#### 5.1.5. Temporal Robustness Evaluation (TRE)

In an image or video sequence, the tracking algorithm starts tracking from different frames as the starting frame (such as tracking from frame 1, 15 and 30 respectively), and the bounding box used for initialization is the ground-truth labeled on the corresponding frame. We took the average of these results to get the TRE result.

#### 5.1.6. Spatial Robustness Evaluation (SRE)

The tracking algorithm may be affected due to the possible error of manually labeled ground-truth. Therefore, in order to reduce this loss, the bounding box was produced by slight displacement of ground-truth and scaling. The displacement is 10% of the target object size, and the scale range is 80% to 120% of the ground-truth, increased by 10% each time. Finally, we took the average of these results as SRE results.

### 5.2. Overall Performance

We compared and evaluated the proposed algorithm with 12 latest trackers. These trackers can be broadly divided into the following three categories:Deep learning tracker (DLT) [10];Correlation filter trackers, including THCF [36], CSK [30], STC [49] and KCF [31];Representative tracking algorithms using single or multiple online classifiers, including MIL [20], Struck [25], CT [50], LSHT [51], TLD [23], SCM [52], MEEM [24].

In Table 1, we quantitatively compared the distance precision at 20 pixels, and the overlap success rate, center location error, tracking speed at 0.5 pixels. Here we list the results of the OTB-50 video sequence [26] and all the OTB-100 video sequences [15]. As we can see from Table 1, the algorithms proposed in this paper were better than the latest trackers in terms of OS rate and CLE, but not as good as THCF [36] in terms of DP rate. The CLE of our method reached 21.2 pixels, and in this area, the THCF [36] tracker got 24.3 pixels as the second-best result. Our tracker ran at about 10 frames per second. At the same time, it can be seen that the DP rate of our method was slightly lower than THCF, because our method took less time and the speed was fast (processing time). Generally, when the tracking accuracy was higher, the time cost was slightly higher. Forward feature extraction took up most of our computing resources (approximately 45% of the computational time per frame). It should be noted that the OTB-100 was more challenging for the entire test sequence, where none of the compared trackers performed as well as the OTB-50. In the latest tracker, the MEEM [24] approach produced third best results on both the dataset OTB-100 and the dataset OTB-100.

Figure 5 and Figure 6 use DP rates and OS rates on data sets OTB-50 and OTB-100 respectively to provide the evaluation results under the OPE (one-pass evaluation), TRE (temporal robustness evaluation), and SRE (spatial robustness evaluation) criteria. In general, the algorithm was superior to the latest methods in three indicators: OPE, TRE and SRE.

As we can see from Figure 5, when OPE, TRE, and STE were used to evaluate DP rate on benchmark *I*, our proposed tracker had similar performance compared to the THCF [36] tracker. The specific data comparison was as follows: 79.1% vs. 73.8%, 78.4% vs. 78.5%, 74.4% vs. 70.7%; in terms of coverage success rate, our tracker and THCF [36] tracker performed slightly betterwith 62.9% and 55.3%, 63.9% and 57.2%, 56.4% and 49.0%, respectively.

Figure 6 shows the similar advantages of the tracker algorithm proposed in this paper on the data set OTB-100: in the evaluation of OPE, TRE and SRE, the tracker proposed by us had better performance than THCF [36] tracker in DP rate, which was 70.4% vs. 67.7%, 69.7% vs. 72.0% and 66.2% vs. 73.0%, respectively. In terms of coverage success rate, the performance of the THCF [36] tracker was comparable to 58.2% and 52.8%, 60.2% and 554.5%, 53.0% and 46.6%.

### 5.3. Complex Scenario Evaluation

We further analyzed the tracking performance of the complex scene video (background clutter, occlusion, scale change) in the data set OTB-100 [15]. Table 2 shows the overlap success rate (%) of all complex scenario OPE evaluations on the OTB-100 dataset, and Table 3 shows its distance precision rate (%).

The following results can be observed from Table 2 and Table 3. First of all, our method effectively dealt with the background clutter, which was realized by the hierarchical semantic and spatial details of CNNs. In contrast, the DLT method useds an unsupervised model to pre-train the network and only took the final output of the trained neural network as the feature. This indicates that CNN features learned under classification supervision (such as VGG-Net) can more effectively identify targets from the background. Secondly, during the process of pre-training models, since we did not completely remove the complex and single semantic information in the final stage, the semantic information was not very sensitive to scale changes. Therefore, when there was a scale change, our method performed well. Finally, our method also had some shortcomings, such as the inability to efficiently deal with tracking failures caused by occlusion and object deformation. This can be attributed to the fact that we did not split the overall feature representation for the time being. In future work, we will consider re-examining modules or component-based models to improve this situation.

### 5.4. Component Modification Comparison Experiment

In this experiment, we split the features, and two algorithms Ours_Deep and Ours_HOG were obtained. Among them, Ours_Deep used the convolution feature for target tracking, and Ours_HOG algorithm only used the HOG feature. In order to prove the effectiveness of multi-feature fusion target tracking proposed by the algorithm in this paper, Ours_Deep, Ours_HOG and other different features integrating tracking algorithms were selected in this experiment for comparative analysis on the OTB-100 test set. Table 4 comparison shows the accuracy and power generation of various tracking algorithms.

### 5.5. Layer Characteristic Difference Analysis

In order to prove the effectiveness of the proposed algorithm in the training of convolutional neural networks and the selection of convolutional layers, we split the convolutional layers on the OTB-100 dataset and tested their different performances respectively. We first tested each convolutional layer involved in the experiment, and then searched and measured the combination of layers 5 and 4. Obviously, we also connected all three layers, 3, 4 and 5 together, as hypercolumns that world be used in the experiment [50]. However, this connection broke the hierarchy at the CNN level and resulted in poor performances in the tracking process. Furthermore, we also tested on AlexNet [55] by using the same scheme. Table 5 shows the use of OPE to perform 10 methods with various combinations, where the DP accuracy value was based on a 20-pixel threshold and the OS accuracy value via the area under the curve (AUC). It can be seen from the experimental results that the experiments based on VGG-Net [40] were more effective than the ones based on AlexNet. When the enhanced semantics had a deeper architecture, they were less sensitive to significant changes in appearance. In addition, the algorithm proposed in this paper used multi-layer CNN to carry out hierarchical reasoning for displacement clues, which improved the tracking performance.

### 5.6. Evaluation Results Analysis

Figure 7 and Figure 8 show the tracking results of 8 complex scene sequences processed by several most commonly used tracking algorithms like THCF [36], MEEM [24], KCF [31], DLT [10]. THCF method works similarly with our method but it was easy to lose the target when the illumination changed suddenly. And when the background clutter had a greater impact on the target, tracking drift happened. MEEM trackers performed well in shape-shifting, rotation, and occlusion sequences, but failed when background clutter and fast motion occurred, because the quantized color channel feature was less effective at processing clutter backgrounds. We also noticed that because MEEM only used the brightness intensity feature in the FreetMan4 sequence, it tended to drift when tracking targets.

The KCF tracker learned the nucleated correlation filter with Gaussian kernel on the HOG feature. It performed well in partial deformation and fast motion sequences, but drifted when the target object was heavily obscured and rotated. DLT approach did not take full advantage of the semantic and fine-grained information and therefore could not track targets on the selected challenging sequence. Struck method was not effective in deformation sequence, background clutter, rotation, and severe occlusion, because Struck, although using structured output to effectively alleviate the sampling ambiguity problem, was not able to effectively deal with large appearance changes due to its manual characteristics.

There are two main reasons for the good performance of our algorithm. First, the algorithm combined the HOG feature and the hierarchical convolution feature, which was more efficient than the traditional manual feature method. Our depth features came from multiple layers of CNN, including the semantics and fine-grained details at the category level, which handled the appearance deformations caused by deformation, rotation, and background clutter. The experimental results show that for the basketball sequence in complex scenes, the 12 latest methods could not track the target well, while our method could achieve a distance precision of 91.8%. Second, the strategic update of memory filter could also deal with the appearance change of tracking target so as to achieve the purpose of automatic recovery of tracking failure.

## 6. Conclusions

Since the hand-designed feature extraction has a great influence on the performance of target tracking, this paper studies how to effectively combine the neural network layered features and HOG features to improve the performance of the object tracking algorithm. In a typical deep neural network, the front-end network layer has more image texture information and has higher resolution, which can accurately locate the target; while the back-end network layer retains more image semantic information, suitable for the target for large-scale positioning. This algorithm proposes the use of deep convolutional neural network hierarchical features and related filtering algorithms, using high-level abstract information to roughly locate the target in the first place, and then using shallow texture details for fine positioning, and finally weighting fusion with HOG features to obtain the target location. This paper proposes that this adaptive weighted feature fusion tracking algorithm combines the advantages of deep learning layered features, HOG features, and related filters, which has achieved good results on a large-scale object tracking test set. The speed is also ahead of the current other tracking algorithms based on deep learning.

## Figures and Tables

**Figure 1 sensors-20-03370-f001:**
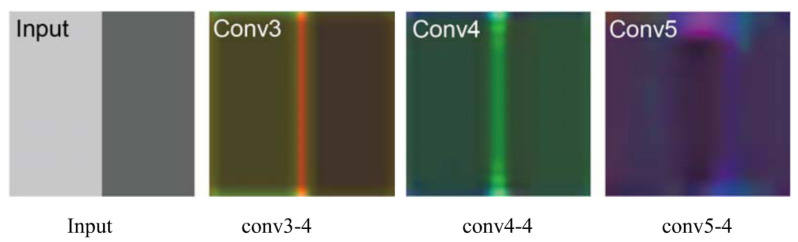
The differential representation of semantic information between CNN layers. It can be seen that the features of the fifth convolutional layer can hardly see the boundary difference with the continuous decrease of spatial resolution, while the features of the third layer show better performance.

**Figure 2 sensors-20-03370-f002:**
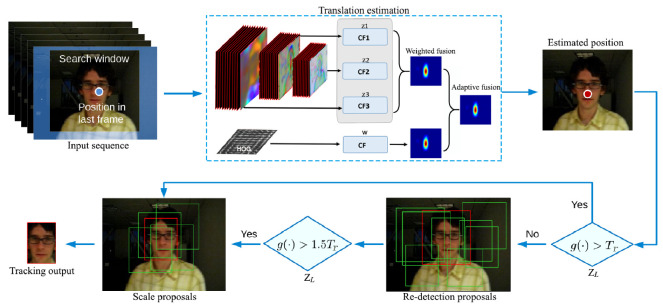
Pipeline of the proposed algorithm.

**Figure 3 sensors-20-03370-f003:**
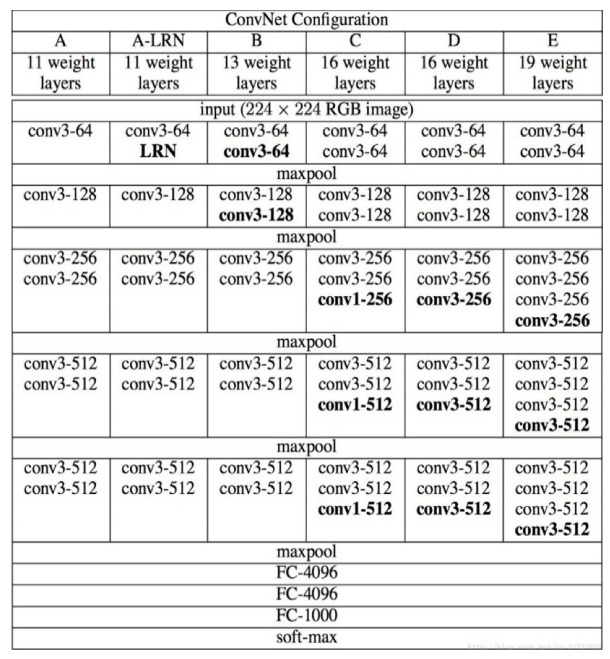
ConvNet configurations.

**Figure 4 sensors-20-03370-f004:**
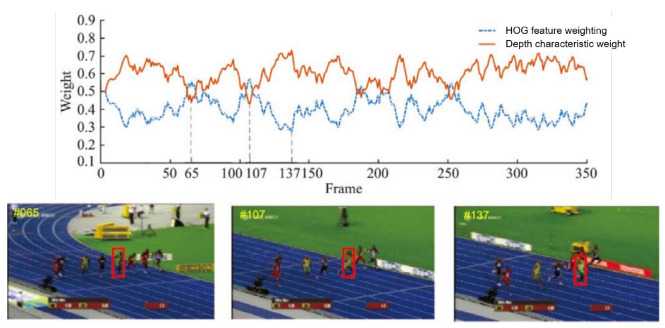
Variation curve of fusion weights of different features on video sequence of Bolt running.

**Figure 5 sensors-20-03370-f005:**
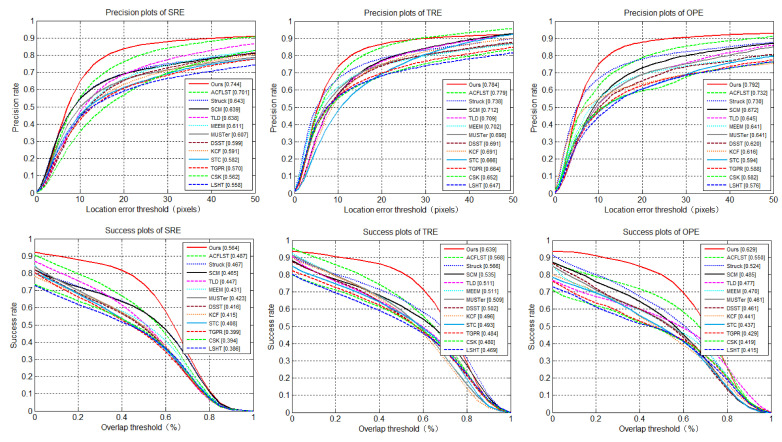
Comparison of distance accuracy rate and overlap success rate using One-Pass Evaluation (OPE), Temporal Robustness Evaluation (TRE) and Spatial Robustness Evaluation (SRE) on OTB-50.

**Figure 6 sensors-20-03370-f006:**
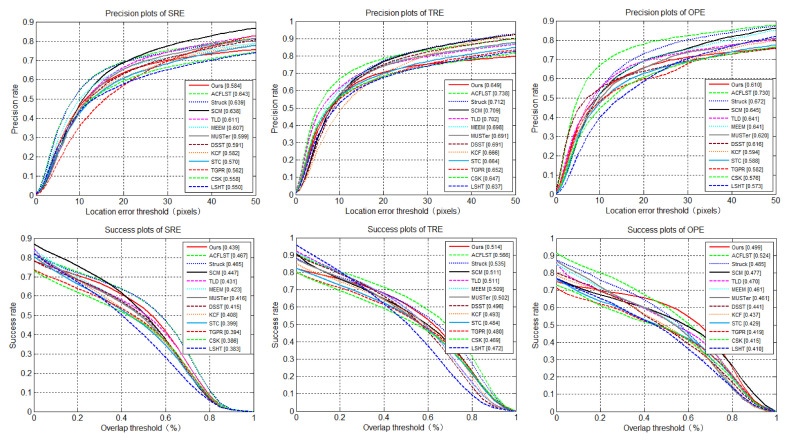
Comparison of distance accuracy rate and overlap success rate using OPE, TRE and SRE on OTB-100.

**Figure 7 sensors-20-03370-f007:**
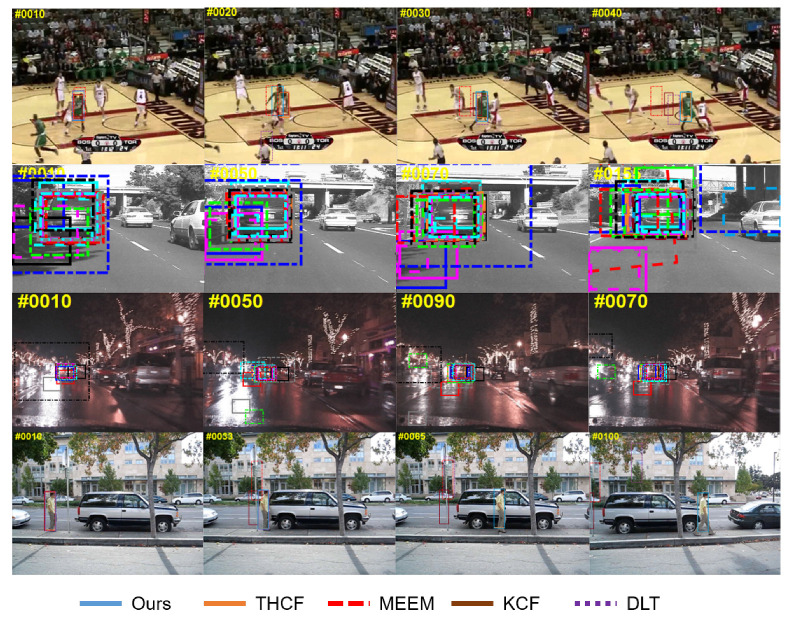
Contrast experiment of this algorithm on multiple challenging video sequences (basketball, car, carDark, david).

**Figure 8 sensors-20-03370-f008:**
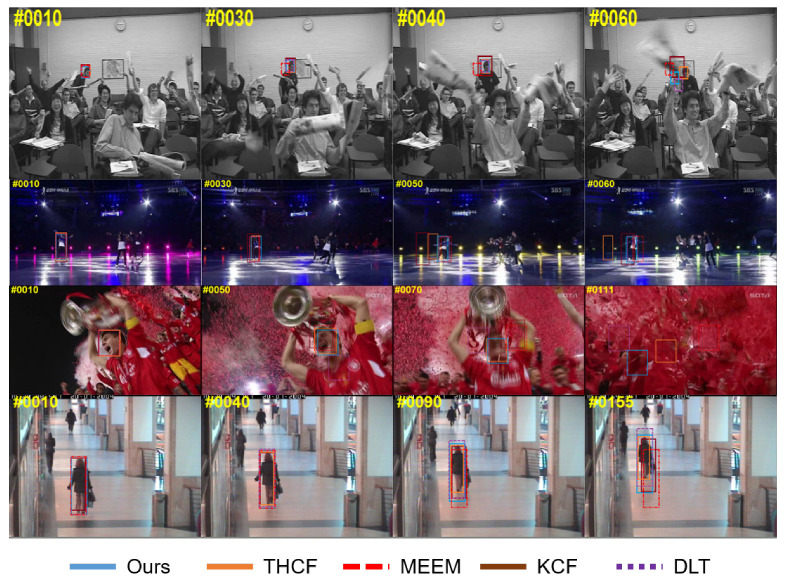
Contrast experiment of this algorithm on multiple challenging video sequences (freeman, skating, soccer, walking).

**Table 1 sensors-20-03370-t001:** Experimental results of proposed method compared with existing methods on the OTB-50 [I] and OTB-100 [II] database.

		Ours	THCF	MEEM	DLT	KCF	STC	Struck	SCM	CT	LSHT	CSK	MIL	TLD
			[36]	[24]	[10]	[31]	[49]	[25]	[52]	[50]	[51]	[30]	[20]	[23]
DP rate	I	79.2	83	79.4	54.8	74.1	54.7	65.6	64.9	40.6	56.1	54.5	47.5	60.8
(%)	II	71.6	78.1	71.3	52.6	69.2	50.7	63.5	57.2	35.9	49.7	51.6	43.9	59.2
OS rate	I	70.3	69.6	68.6	47.8	62.2	36.5	55.9	61.6	34.1	45.7	44.3	37.3	52.1
(%)	II	63.5	62.2	62.1	43	54.8	31.4	51.6	51.2	27.8	38.8	41.3	33.1	49.7
CLE	I	18.7	20.9	25.5	65.2	35.5	80.5	50.6	54.1	78.9	55.7	88.8	62.3	48.1
(pixel)	II	23.8	27.7	29.8	66.5	45	86.2	47.1	61.6	80.1	68.2	305	72.1	60
Speed	I	13.5	12.8	20.8	8.59	245	687	10	0.37	38.8	39.6	269	28.1	21.7
(FPS)	II	11.4	11.8	20.8	8.43	243	653	9.84	0.36	44.4	39.9	248	28	23.3

**Table 2 sensors-20-03370-t002:** Overlap success (%) in terms of comples scene on the OTB-100 dataset.

Attributes	Ours	THCF [36]	MEEM [24]	DLT [10]	KCF [31]	STC [49]	Struck [25]	SCM [52]	CT [50]	LSHT [51]	CSK [30]	MIL [20]	TLD [23]
lllumination variation(23)	72.0	72.1	70.2	61.7	58.3	57.5	56.5	34.4	41.1	47.4	53.9	28.0	43.2
Out-of-plane rotation(37)	78.6	74.7	73.1	66.4	61.2	62.1	58.2	36.9	43.5	50.4	57.8	35.1	45.7
Scalvariation (28)	67.5	69.5	68.8	56.9	47.2	46.3	50.8	32.8	34.0	47.9	61.7	34.5	48.7
Occlusion(27)	77.4	76.9	73.8	65.9	61.4	65.1	57.5	31.0	41.0	49.6	58.3	36.4	41.6
Deforrnation(17)	85.7	83.7	78.8	63.5	65.9	73.4	60.3	31.9	39.5	50.4	55.7	44.2	40.7
Motionblur(12)	65.8	66.4	66.7	65.5	54.8	58.0	52.4	21.3	34.3	50.0	32.1	23.4	48.2
Fastmotion(17)	67.6	67.3	65.7	67.4	49.5	55.1	49.7	21.4	36.7	55.1	32.6	32.7	45.9
In-plane rotation(31)	76.2	73.8	67.7	63.7	58.4	60.4	65.1	37.9	45.3	53.7	54.8	32.5	45.6
Outofview(6)	69.3	68.7	68.0	74.0	54.4	63.6	53.7	30.0	40.8	55.9	44.4	42.3	50.9
Background clutter(21)	75.5	75.4	75.5	72.1	66.5	67.9	59.7	37.7	47.9	53.5	53.6	40.3	39.8
Low resolution (4)	45.1	46.6	42.7	37.6	28.7	26.8	32.0	28.7	28.2	24.8	54.0	14.4	33.6
Weighted Average	73.9	73	70.6	64.3	57.6	59.8	56.7	32.9	40.8	50.5	53.2	34.3	44.5

**Table 3 sensors-20-03370-t003:** Distance precision (%) in terms of comples scene on the OTB-100 dataset.

Attributes	Ours	THCF [36]	MEEM [24]	DLT [10]	KCF [31]	STC [49]	Struck [25]	SCM [52]	CT [50]	LSHT [51]	CSK [30]	MIL [20]	TLD [23]
lllumination variation (23)	79.1	79.8	78.5	74.4	63.8	69.9	72.7	57.8	46.6	55.7	56.2	32.0	50.1
Out-of-plane rotation (37)	83.6	84.1	84.1	83.3	69.8	77.2	76.0	57.2	55.4	61.3	62.3	49.3	58.4
Scal variation (28)	76.7	84.5	82.5	77.6	62.6	68.6	75.7	56.5	49.9	65.0	66.3	47.5	60.5
Occlusion (27)	84.0	84.1	83.1	80.5	67.6	80.2	77.0	53.4	52.9	59.1	65.2	44.1	54.3
Deforrnation (17)	86.6	86.7	85.0	84.3	71.4	82.0	71.6	51.0	50.0	57.2	57.5	48.7	48.3
Motion blur (12)	67.6	68.6	70.5	72.1	54.2	66.2	59.9	32.3	35.2	55.5	35.2	34.8	51.0
Fast motion (17)	66.7	71.7	70.0	74.9	49.0	60.6	58.1	28.8	37.2	60.5	33.4	39.5	55.0
In-plane rotation (31)	80.8	81.6	79.7	81.1	69.0	73.2	78.2	53.1	54.6	63.5	60.5	45.2	60.0
Out of view (6)	73.5	72.8	72.7	72.3	51.0	66.3	53.3	41.2	37.2	55.3	44.5	39.2	57.3
Background clutter(21)	81.2	82.0	83.7	80.6	72.9	75.9	69.6	53.5	58.2	59.9	59.3	45.2	41.9
Low resolution (4)	72.4	75.7	76.4	90.4	44.0	63.3	74.0	50.9	46.4	55.3	66.0	32.5	56.4
Weighted average	79.3	81.1	80.4	79.4	64.9	73.0	72.2	51.3	50.1	60.1	57.5	43.6	54.5

**Table 4 sensors-20-03370-t004:** Performance evaluation using different features.

	Ours	Ours_Deep	Ours_HOG	SAMF [53]	KCF [31]	DSST [54]	CN
OS(%)	64.3	60.9	59.3	57.8	55.7	52.9	49.3
DP(%)	85.8	85.4	72.2	70.7	68.2	67.1	60.2

**Table 5 sensors-20-03370-t005:** Evaluation of experiments using multiple convolutional layers on the OTB-100 dataset.

	Ours	Vgg-Net-5	VggNet-4	VggNet-3	VggNet-5-4	VggNet-543	AlexNet-5	AlexNet-5-4-3	AlexNet-5-4	AlexNet-543
OS(%)	64.3	56.7	54.9	52.4	55.7	52.2	49.5	47.5	47.2	46.9
DP(%)	85.8	84.7	78.3	72.5	84.8	77.4	67.4	66.5	65.9	65.7

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
