# Peer review of "Visual Tracking via Deep Feature Fusion and Correlation Filters"

_sensors, 2020, doi:10.3390/s20123370_

Round 1

Reviewer 1 Report

Authors proposed a correlation based trackers that exploits deep features and hog features to compute the target model.

Authors did not clearly defined their contribution. The authors exploited deep features as well as hog features to exploit target information which is already proposed. I suggest authors to read these papers and improve their method.

1]- Ma, Chao, et al. "Hierarchical convolutional features for visual tracking." Proceedings of the IEEE international conference on computer vision. 2015.

2]- Ma, Chao, et al. "Robust visual tracking via hierarchical convolutional features." IEEE transactions on pattern analysis and machine intelligence 41.11 (2018): 2709-2723.

3]- Fiaz, Mustansar, et al. "Handcrafted and deep trackers: Recent visual object tracking approaches and trends." ACM Computing Surveys (CSUR) 52.2 (2019): 1-44. 

Paper needs an extensive review to remove grammatical errors.

Reviewer 2 Report

This paper proposed a visual tracking algorithm based on CNN feature and correlation filters.  This paper should be re-written in many parts. With the current version, there are many typos and mis-expressions. I have the following comments to improve this mansucript:

  • In Abstract, there is "In this thesis,~". This is not thesis, nut a technical paper.  
  • In Section 2.1, it would be better to add some fuzzy-based tracking schemes to give more good understanding as:
        - A fuzzy inference approach to template-based visual tracking, Machine Vision and Applications 23(3):427-439, DOI: 10.1007/s00138-010-0314-8, May 2012.

       - Novel Target Segmentation and Tracking Based on Fuzzy Membership Distribution for Vision-Based Target Tracking System, Image and Vision Computing, volume 24, 2006. 

       - Fuzzy logic approach to visual multi-object tracking, Neurocomputing

Volume 281, 15 March 2018, Pages 139-151.  
  • In p. 4, please correct "Set T_r as threshold to determine if ~".
  • In p. 5, where is alpha? There is no use of alpha in Eq. (2). Also, eta was not defined anywhere. In my opinion, this may be another weighting factor.
  • In p. 6, Authors mentioned "As shown in figure 3.2 ~". But there is no figure 3.2. I think this is Figure 3. There are so many typos like this. Authors should re-check and correct typos very carefully.
  • In Section 4.4, this section was very difficult to understand the evaluation because of many breaking the connection between equations. In Eq. (11), authors defined a_l. How to determine this parameter? What authors want to determine in this section?  Very confusing what they want.
  • In Experimental Results section, authors showed the performance of the proposed method. The performance indices have been well defined and evaluated.  I think it would be better to show the processing time (inference time) and complexity analysis. 

I think this paper should be revised significantly to improve the quality of it.

Reviewer 3 Report

This paper combines classical target tracking with deep learning to obtain an improved tracker. The ideas seem to me original and very interesting and results look good, and it deserves an oportunity to be published. However, presentation is under standards. First, the manuscript has to be thoroughly checked for English grammar and typos. Second, the equations have to be checked for notation coherence, and all variables and parameters have to be defined first time they appear. Third, the details of the method that are irrelevant for the flow of the narrative can be encapsulated in specific sections or put into an appendix. This is, follow a top-down approach, and make use of graphics explanation as you do for instance in Figure 2. Finally, to be fair in the comparisons, you should include in the video testing, Fig.7&8, the THCF method, as it seems the one that gives the nearest results to yours. Details follow:

-Eq. 1 has inconsistent notation with subindexes and superindexes, Eq. 12 and Eq. 13 shows C both uppercase and lower case, in Eq. 8 and 10 w is in uppercase and lowercase, p is both upper and lower case in Algorithm 1, in Eq. 5 and 7 x is both uppercase and lowercase, and subindexes are used as wxh and w,h

-Eq. 2 is incomplete as there seems to lack the function D

-parameters are undefined, as \alpha in line 143, \lambda in Eq. 8, only defined much later, as regularization parameter

-vector i, or vector x? in line 185

-where is figure 3.2?

-Intersection over Union (IoU) is only defined much later after first mentioned

-Legends in Fig 5 and 6 are not readable

-HOG (Histogram of Oriented Gradients) is only given as acronym in the paper please define and refer bibliography

-then we make a solid space, line 249: What does it mean?

-these objects represent that a target object that has not been set in advance can be well captured for model-free tracking: What does it mean?

Round 2

Reviewer 1 Report

Authors have improved the previous version but still there are many grammar and typos.  Also there are still issues such as:

  • Authors compared their approach with old methods as, tracking is a growing field.
  • What do they mean by low time complexity?
  • Line 73, what do they mean manual features?
  • line298. IOU is a measure parameter not threshold.
  • line 56. check the reference.

Reviewer 2 Report

This paper has been well revised based on my comments. However, still there are some typos like" Finally, we conducted a large number of experiments on benchmark data sets OBT-100 and OBT-50, and confirmed that our proposed algorithm is effective." should be corrected as " Finally, we conduct a large number of experiments on benchmark data sets OBT-100 and OBT-50, and confirm that our proposed algorithm is effective."  Therefore, authors should re-check on typos carefully before submitting the final manuscript.

Reviewer 3 Report

This version is much improved with respect to the previous version. Authors also included video shots comparison with THCF method. However, many English grammar and typos remain. I list some of them below. Also, there are still some issues with equations, see also below. Other issues are:

-Authors speak of low time complexity of their method. Do they refer to time cost or to time complexity? If it is complexity I didn't see any complexity study of the method and/or of the other ones in the paper, be empirical o theoretical. Lower time costs do not necessarily mean lower time complexity and viceversa, only mean it in the long run.

-line 92: “manual features”: what are manual features in object tracking?

-“A given threshold is also called Intersection over Union (IoU)”: What does it mean? That IoU is used as one of the thesholds? Please change the sentence.

-Distance Precise: I think it should be Distance Precision

-“the 12 latest trackers” : I would use “12 latest trackers” or 12 recent, without the article “the”, because the authors might miss some recently published tracker.

-the authors say: “We set the regularization parameter to \lambda = 10^-4 of Equ. 7 and make 0.1 kernel bandwidth”, did the authors obtain these parameters values by trial and error? Are they valid for all scenes used?

-“in the foregoing time t”:  foregoing means preceding, or previous. But you use t-1 and t, and t-1 precedes t, t follows t-1.

-In formulae 1 and 2: what does the superindex “I” mean? Should it appear in the t-1 time too?

-line 214: “Our algorithm sets output”: what does it mean?

-line 217: variable “x” is used several times in the paper. It is used always with the same meaning? If not, please use different variable name when the meaning is different.

-in formula (7): what does the subindex 2 mean at the norm of z?

-in line 223: does superindex T stand for transposed?

-in line 230: dot product: which dot product are you using?

Typos/grammar issues found:

In abstract:

-line 2: it has attracted “a much attention”: drop the “a”

-line 5: “posinge” -> pose

-line 5, “significantly” is repeated

-sentence between line 10 and 11, either comma has to be changed by dot or Rich has to be written lower case. Also, authors use passive when they use active voice elsewhere, please change to active voice.

-line 64: sentence starting by “Learned” has no subject

-line 80: delete “reducing”

-line 89: “proposesd”-> proposed

-in caption in Figure 1: “It can be found” or it can be seen?

-line 156: “bt”: ?

-line 178: “However”: according to English dictionary, however means: used to introduce a statement that contrasts with or seems to contradict something that has been said previously”. But I didn't see any contrast here

-line 186: “so less concerned”: so we are less concerned

-“HOG feature is a traditional artificial design feature”: first and last time that “artificial design” appears. If it is important, add a reference. If it is an error, or typo delete or correct it

-Fig 4 caption: “features on Bolt”: on Bolt what?

-“The formula is these”: Either “the formula is this”, or the formulas are these”

-“As what be seen from the figure”: there is some gram”mar error

-in Algorithm 1: and extracting -> and extract
